# The Effects of Titanium Dioxide Nanoparticles on Osteoblasts Mineralization: A Comparison between 2D and 3D Cell Culture Models

**DOI:** 10.3390/nano13030425

**Published:** 2023-01-20

**Authors:** Gabriela de Souza Castro, Wanderson de Souza, Thais Suelen Mello Lima, Danielle Cabral Bonfim, Jacques Werckmann, Braulio Soares Archanjo, José Mauro Granjeiro, Ana Rosa Ribeiro, Sara Gemini-Piperni

**Affiliations:** 1Postgraduate Program in Odontology, Unigranrio, Duque de Caxias 25071-202, Brazil; 2Directory of Life Sciences Applied Metrology, National Institute of Metrology Quality and Technology, Rio de Janeiro 25250-020, Brazil; 3LabCeR Group, Federal University of Rio de Janeiro (UFRJ), Rio de Janeiro 21941-901, Brazil; 4Visitant Professor at Brazilian Center for Research in Physics, Rio de Janeiro 22290-180, Brazil; 5Materials Metrology Division, National Institute of Quality and Technology, Rio de Janeiro 25250-020, Brazil; 6NanoSafety Group, International Iberian Nanotechnology Laboratory, 4715-330 Braga, Portugal; 7Labεn Group, Federal University of Rio de Janeiro (UFRJ), Rio de Janeiro 21941-901, Brazil

**Keywords:** TiO_2_ NPs, titanium dental implants, 3D spheroids, osteoblasts

## Abstract

Although several studies assess the biological effects of micro and titanium dioxide nanoparticles (TiO_2_ NPs), the literature shows controversial results regarding their effect on bone cell behavior. Studies on the effects of nanoparticles on mammalian cells on two-dimensional (2D) cell cultures display several disadvantages, such as changes in cell morphology, function, and metabolism and fewer cell–cell contacts. This highlights the need to explore the effects of TiO_2_ NPs in more complex 3D environments, to better mimic the bone microenvironment. This study aims to compare the differentiation and mineralized matrix production of human osteoblasts SAOS-2 in a monolayer or 3D models after exposure to different concentrations of TiO_2_ NPs. Nanoparticles were characterized, and their internalization and effects on the SAOS-2 monolayer and 3D spheroid cells were evaluated with morphological analysis. The mineralization of human osteoblasts upon exposure to TiO_2_ NPs was evaluated by alizarin red staining, demonstrating a dose-dependent increase in mineralized matrix in human primary osteoblasts and SAOS-2 both in the monolayer and 3D models. Furthermore, our results reveal that, after high exposure to TiO_2_ NPs, the dose-dependent increase in the bone mineralized matrix in the 3D cells model is higher than in the 2D culture, showing a promising model to test the effect on bone osteointegration.

## 1. Introduction

The development of nanotechnology is increasing exponentially, especially in the production of biomaterials for dental implants [1,2,3]. This market moved around US$10.87 billion in 2021 and could reach US$12.49 billion in 2022 [1,4]. In this scenario, titanium (Ti) is the metallic material most commonly used for implant applications due to its excellent biocompatibility and osteointegration properties [2,5]; however, the failure of dental implants continues to increase [6]. Although the causes are multifactorial and often related to microbial colonization (biofilms), new questions have been raised about the role of corrosion and/or wear process in the progress of implant failure [7]. Recently, Ti-like particles (mainly titanium dioxide nanoparticles (TiO_2_ NPs)) have been found in the peri-implant mucosa and bone cells [6,8,9].

Prothesis degradation processes occur in all material classes, with degradation by-products known to be hazardous. Nonetheless, nanoscale metallic debris released are of tremendous concern since they exhibit enhanced toxicity and dissolving capacities, compared with micron-size polymeric and ceramic debris [7]. Micromovements between the implant abutment and the bone or mucosa unavoidably lead to mechanical wear of the material (formation of nano and microsized debris), which in a corrosive environment result in the release of metallic ions [7,10,11,12]. Biologically, this event is related to macrophage activation [7,10,11,13] and the onset of pro-inflammatory response, with the release of cytokines such as IL-1β, IL-6, and TNF-α in the peri-implant area, which culminates in reduced osteoblastogenesis, increased osteoclastogenesis, and bone loss in the periprosthetic region [11,13]. Bone loss limits the ability of a prosthesis to withstand physiological loads, giving rise to a need for revision surgery.

Many in vitro and in vivo studies have shown that titanium particles induce pro-inflammatory and toxic effects in the peri-implant environment [7,14,15,16,17]. However, some in vitro studies also suggest that TiO_2_ NPs can stimulate bone formation [2,18]. Thus, controversy remains about the actual role of TiO_2_ NPs on bone cells, likely due to the two-dimensional in vitro models used.

It has been widely reported that 2D osteoblast cell cultures [13,14,15,16,17,19] may lose their original tissue organization and polarity and have limited protein–protein interactions [2,18]. They may exhibit integrins and changes in the cytoskeletal organization that alter their original morphology [2,18,20]. In addition, cells grown in a 2D monolayer often exhibit altered metabolism, phenotype, and gene expression, and the interactions between cells and the extracellular matrix are different from in vivo tissues, which have a three-dimensional architecture [2,20]. In contrast, 3D culture models exhibit greater cell–cell and cell–matrix interactions, which are closer to the in vivo model [21]. Moreover, cellular polarity, which is important for cellular organization and functionality, remains unaltered [2,21,22]. Therefore, surface receptors can bind to extracellular matrix proteins, activate cellular biochemical signals, and influence cell proliferation, differentiation, and mineralization [2,21].

Therefore, in this work, the effects of TiO_2_ NPs on the differentiation and production of a mineralized matrix of human osteoblasts cultured in monolayer (2D) and osteoblast-like spheroid culture models (3D) were investigated. Cell viability, morphology, differentiation, mineralization, and nanoparticle internalization were investigated after exposure to NPs. Results demonstrate that TiO_2_ NPs lead to a dose-dependent increase in mineralization, although these TiO_2_ NPs are clinically known for their possible immune system activation.

## 2. Materials and Methods

Titanium anatase dispersion: TiO_2_ NPs (SIGMA, Kanagawa, Japan) with primary particle size < 25 nm and surface area of 45–55 m²/g were suspended in ultrapure water (2 mg/mL; pH 4) and dispersed using a direct ultrasound (Q-Sonica) equipped with a 19 mm tip. The sonication was carried out in an ice bath at 32 W of acoustic delivery power for 15 min with 8 s (pulse mode on) and 2 s (pulse mode off), following a protocol previously described by the group [2,5]. After 24 h of stabilization, particle size and particle agglomeration (zeta potential analysis (ζ (mV)) and the polydispersion index (PdI)) were determined by dynamic light scattering (DLS, Zeta-Sizer Nano ZS, Malvern Instruments GmbH, Malvern, UK). DLS measurements were performed at 25 °C using 10 mm polystyrene disposable cuvettes.

To confirm particle size in the medium culture, titanium particles were alternatively suspended in high glucose Dulbecco’s Modified Eagle Medium (DMEM, Gibco, Waltham, MA, USA) supplemented with 10% fetal bovine serum (FBS, Gibco) and 1 mg/mL bovine serum albumin (BSA; Sigma-Aldrich, St. Louis, MO, USA) to avoid particle re-agglomeration.

Cell culture: The human osteoblast cell line (SAOS-2) was supplied by the Cell Bank of Rio de Janeiro (BCRJ, Rio de Janeiro, Brazil) packed in frozen ampoules and kept in liquid nitrogen. Cells were thawed and expanded into cell culture flasks (Corning) with DMEM medium supplemented with 10% FBS and 1% penicillin/streptomycin (PS—10,000 units/mL of penicillin and 10,000 μg/mL of streptomycin) (PS, Gibco) in a humidified incubator (5% CO_2_, 37 °C). Cell contamination with bacteria, fungi, or mycoplasma was analyzed as previously reported [2,5]. For the 2D model, 10,000 cells/well were seeded in standard flat-bottom 96-well plates for 24 h.

3D culture: For 3D spheroid formation, 96-well U-bottom plates (Corning, Corning, NY, USA) were coated with a thin layer of 1% ultrapure agarose (Sigma-Aldrich), and 10,000 cells were seeded in each well in 200 μL DMEM high glucose medium supplemented with 10% FBS and 1% PS and then incubated for 3 days. Cell growth, shape, and morphology were analyzed on an inverted optical microscope (Nikon Eclipse, Tokyo, Japan), following a protocol previously described [2].

3D cell cytoskeleton: Spheroids were washed with 0.01 M PBS and then fixed with 4% paraformaldehyde (PFA). The cell membrane was then permeabilized by incubation for 2 h with 0.2% BSA + 0.1% Triton X-100 in 0.01 M PBS for 2 h at room temperature (RT). The spheroids were washed three times with a blocking solution containing 50 nM NH_4_Cl in 0.01 M PBS. Phalloidin solution (500 ng/mL, stock 1:40, ThermoFisher Lot: F432) diluted in 0.2% BSA + 0.1% Triton X-100 in 0.01 M PBS was then added and incubated for 2 h, and later an additional 30 min with DAPI (1:500) (SIGMA-ALDRICH—Lot: 583-93-7). Cell morphology was visualized using a confocal fluorescence microscope (DMI 6000, Leica, Teaneck, NJ, USA).

NPs exposition: Both 2D and 3D cell cultures were exposed to 0, 5, and 100 μg/mL TiO_2_ NPs suspended in incomplete osteogenic medium composed of DMEM supplemented with 10% FBS, 50 μg/mL of ascorbic acid (Sigma), 100 Mm of β-glycerophosphate (Sigma), and antibiotics for 3 and 21 days. Cells without TiO_2_ NPs treatment were used as control.

Cytotoxicity assay: After NPs exposition, the cells were washed three times with 0.01 M PBS and then incubated with 0.125% Trypsin (kept in a humidified incubator with 5% CO_2_, 37 °C) for 5 min. Trypsin was blocked by adding culture medium with 10% FBS, and 2D adherent cells and 3D spheroids were mechanically dissociated. The cells were centrifuged for 7 min at 500 x g (4 °C), and the pellet was resuspended in 100 μL annexin-binding buffer (Dead Cell Apoptosis Kit for Annexin V; Kit Life and Dead, Life Technologies). The samples were incubated for 15 min (RT) in 3 μL annexin/fluorescein (FITC) solution and 1 μL propidium iodide (according to the manufacturer’s instructions). All analyses were performed in a flow cytometer (FACSAria III, BD Biosciences, Franklin Lakes, NJ, USA).

Morphology analysis: Cells were washed with 0.01 M PBS and processed for scanning (SEM) and transmission (TEM) electron microscopy. Briefly, cells were fixed using modified Karnovsky (2% paraformaldehyde, 2.5% glutaraldehyde in 0.1 M sodium cacodylate buffer, pH 7.2) for 2 h at RT and washed with 0.1 M cacodylate buffer. The samples were post-fixed with 1% osmium tetroxide in cacodylate buffer (1:1) for 30 min in the dark, then washed with cacodylate buffer and dehydrated in ethanol (VETEC—1567). Then, for TEM analysis, samples were contrasted in a bloc with 1% of uranyl acetate, dehydrated in acetone, and embedded in Spurr. Ultra-thin sections were also analyzed using EDS in scanning transmission electron microscopy (STEM) mode in a TITAN 80–300 electron microscope (FEI, Netherlands (300 kV).

Alternatively, for SEM analysis, the cells were dried at a critical point (Autosamdri^®^-815, Series A) and metalized with gold (in a current of 40 mA for 90 sec). The 2D cell samples were analyzed in a scanning electron microscope (JEOL Field Emission Gun-JSM-7401F) with an acceleration voltage of 1 kV. The 3D cells were analyzed under a helium ion beam microscope (HIM) (Carl Zeiss Orion Nanofab—beam current of 0.8 pA, using an electron flood gun to compensate for the positive charge).

Differentiation and analysis of the cell matrix: Cell differentiation was evaluated by alkaline phosphatase histochemistry. The cells were cultured at different times (3, 7, and 21 days). The alkaline phosphatase labeling kit (Sigma-Aldrich Lot: APF-1KT) was used, which is based on the application of 500 µL of diazonium and naphthol salt solution for 30 min in the dark. Afterward, the reaction was stopped with tridistilled water. Positive cells marked in red were photographed under an inverted optical microscope (Nikon Eclipse TS100), using the photo program (Leica Applications Suites—LAS EZ). 

To evaluate the production of the mineralized matrix, alizarin red staining was performed after 3, 7, and 21 days of culture. Cells were fixed with 4% PFA and exposed to 1% alizarin red solution (Sigma-Aldrich) at RT for 30 min, then rinsed with ultrapure water. To quantify matrix mineralization, alizarin red-positive nodules were dissolved in a solution of 0.5 N HCl with 5% SDS. The optical density (OD) values of absorbance were quantified spectrophotometrically at a wavelength of 450 nm using a microplate reader (Biotek Synergy 2 multi-mode detection with gen5 software).

Statistical analysis: Data were presented as mean ± standard deviation (SD). The Gaussian distribution of the samples was tested, and the statistical significance of the data was evaluated using one-way ANOVA or unpaired *t*-tests. The *p* values are shown in the figures and statistical significance was considered when *p* < 0.05. Each experiment was performed three times, with triplicates.

## 3. Results

### 3.1. Characterization of TiO_2_ NPs

TiO_2_ NPs with a primary size of 25 nm were used to mimic the wear particles released by dental implants. The physicochemical characterization of the primary TiO_2_ NPs was already published [2,5]. TEM micrographs revealed that TiO_2_ NPs in ultrapure water were agglomerated, requiring the implementation of a dispersion protocol (Figure 1A). Dark-field STEM images show the morphology and agglomeration of TiO_2_ NPs after dispersion (direct probe sonication), and the STEM/EDS Ti-K map (in blue) confirmed the identity of the TiO_2_ NPs (Figure 1B). DLS analysis (Figure 1C) showed that the mean diameter (DH (nm)) of the TiO_2_ NPs was 135 ± 24 nm in water and increased significantly (*p* < 0.05, unpaired *t*-test) in cell culture medium (156 ± 14 nm), maintaining a polydispersion index (PdI) of less than 0.2. Finally, the zeta potential analysis (ζ (mV)) in water and culture medium showed a significant decrease in the zeta potential value after medium contact, indicating the formation of protein and ionic corona on TiO_2_ NPs surface (*p* < 0.05, unpaired *t*-test) (Figure 1C).

### 3.2. Effect of TiO_2_ NPs on the Morphology of 2D and 3D Human Osteoblasts

In this study, human osteoblasts (SAOS-2) were cultured as monolayers or spheroids. After 72 h of seeding, cells were exposed for 72 h to 100 µg/mL of TiO_2_ NPs. Optical microscopy images show the conventional SAOS-2 morphology (Figure 2A, left panel). In monolayers, the cells exhibit an epithelial-like phenotype, which is maintained after exposure to TiO_2_ NPs. SAOS-2 spheroids have a round shape with a well-organized cytoskeleton (Figure 2B,C), also maintaining their morphology upon titanium exposure (Figure 2C). However, a 29% increase (*p* = 0.0151, unpaired *t*-test) in diameter and volume was observed after exposition to TiO_2_ NPs (Figure 2D).

To confirm whether ultrastructural changes occurred after treatment with TiO_2_ NPs, scanning electron microscopy (SEM) analysis was performed and showed that SAOS-2 in 2D and spheroids (3D) maintained their morphology after 3 days of exposure to TiO_2_, without changes in their cell–cell contact (Figure 3A,B). Moreover, SEM-EDS analysis confirmed the presence of TiO_2_ NPs on the surface of both cell models. A detail of interaction of TiO_2_ NPs with spheroids (Ti-k, marked in blue) can be observed in Figure 3C.

### 3.3. Effect of TiO_2_ NPs on Human Osteoblast Viability

Flow cytometry analysis with PI/annexin after 3 and 21 days of culture did not show TiO_2_ NPs cytotoxicity, both in the monolayer (Figure 4A) and in the spheroids models (Figure 4B). The levels of apoptosis and necrosis were similar in all conditions evaluated.

### 3.4. Internalization of TiO_2_ NPs in 2D and 3D Culture of Human Osteoblasts

Transmission electron microscopy (TEM) showed, both in 2D and 3D models, the internalization of TiO_2_ NPs, that preferentially located in the cell cytoplasm within membrane-like-vesicles or after cell-membrane disruption, possibly in multivesicular bodies (MVBs) or auto-phagolysosomes delimitated by the membrane (Figure 5). 

### 3.5. Differentiation and Mineralization of Human Osteoblasts after Exposure to TiO_2_ NPs

To understand the influence of TiO_2_ NPs on the differentiation and mineralization of both cell models (2D and 3D), analyses of alkaline phosphatase (ALP) (differentiation marker) and alizarin red (mineralization marker) were performed. For these analyses, osteoblasts were cultured for up to 14 days, and two exposure concentrations (5 and 100 µg/mL) of TiO_2_ NPs were used. Previous data in human primary osteoblasts 2D histochemical micrographs showed that the treatment of TiO_2_ NPs did not enhance the labeling for ALP (marked in red) after 14 days of culture (Appendix A). However, in the mineralization analysis, there was a dose-dependent increase in alizarin staining after 14 days of treatment with 100 µg/mL (marked in intense red) compared to the control (Appendix A).

To compare differences in mineralization occurring in 2D and 3D models, we performed alizarin staining after 3, 7, and 14 days after 5 μg/mL or 100 μg/mL TiO_2_ NPs exposure in both models. Alizarin red results showed a significant dose-dependent increase in mineralization at 14 days compared to the control, both in 2D (Figure 6A) and 3D (Figure 6B). Moreover, when treatment values are normalized by control values, the mineralization increase is higher in the 3D model when compared with the 2D model, suggesting that both models can present different results in the mineralization evaluation (Figure 6C and representative images in Figure 6D).

## 4. Discussion

Titanium is the main material employed in the dental implant industry, due to its high mechanical strength, low elastic modulus, corrosion resistance, ductility, and biocompatibility [6,9]. However, tribocorrosion processes at the implant surface lead to accelerated bone loss, compromising osseointegration, and increasing periprosthetic failure [2,5,6,7,8,9,23,24]. The hostile electrolytic environment (oxidation/reduction) together with mechanical action at the interface enables the tribocorrosion phenomena [7,10]. As a consequence, degradation products (released from implants) including metal ions, micrometric, and/or nanometric metallic debris (TiO_2_ NPs) can be internalized by cells in the bone niche, possibly generating cytotoxic effects [6,9,10]. The adverse effects of TiO_2_ NPs vary widely in the literature, which raises concern among authorities and physicians due to their high prevalence [5,10,17]. Literature data reveal that inflammatory stimuli associated with cytokine overproduction and increased production of reactive oxygen species are referred to as primary toxic effects that lead to cell death [6,9,13,17].

Some authors explained that this mechanism leads to activation of immunological sentinels and accumulation of antigens such as ions, nanoparticles, microparticles, and bacterial antigens via the functional interface between dental implant and tissue. This leads to immunological cell polarization and follows dental implant loss [6,9].

Most available studies that evaluate osteoblast response to TiO_2_ NPs use 2D cell culture models, which have shown limitations regarding cell growth and cell–cell and cell–matrix interactions, among others [25,26,27,28]. Few studies evaluate the influence of TiO_2_ NPs on the physiology of bone cells grown in 3D models such as spheroids [2]. Osteoblast spheroids can be considered as a culture model that better mimics living cells in terms of structural and biofunctional properties and provides more reliable results compared to conventional 2D cell cultures (Figure 7) [2,25,29]. Despite this, there are some limitations to spheroid culture, mainly because cellular environments are not similarly exposed to the culture medium. This can lead to the formation of a microenvironment inside the spheroids that can select groups of cells [30,31]. Partial diffusion of nutrients or oxygen can induce necrotic areas in the central area of the spheroids [32]. However, well-characterized multicellular spheroids exhibit different levels of extracellular matrix deposition, growth factor secretion, and gene expression profiles [2]. The viability, morphology, and gene expression of osteoblastic spheroids are contact-dependent, and single or co-culture spheroids have been shown to have an impact on bone cell function [33]. Interestingly, a study reported that primary osteoblasts and pre-osteoblasts MC3T3-E1 can differentiate into osteocytes when grown in 3D cultures [34]. Therefore, 3D culture models can be used to study the pathophysiological reactions of TiO_2_ NPs in bone metabolism compared to 2D cultures. A previous study by W. Souza et al., on the cytotoxicity effect of TiO_2_ NPs on osteoblast spheroids, revealed that 72 h exposition to TiO_2_ NPs can alter the cell cycle, without interfering with osteoblasts’ ability to differentiate and mineralize and significantly increase collagen and pro-inflammatory cytokine secretion [2]. In the present study, a longer exposure period (21 days) was assessed to compare 2D with 3D osteoblasts models to better understand their relevance for nanotoxicological studies.

TiO_2_ NPs are chemically stable, have antibacterial properties, and induce less toxicity than other nanostructures, and, when exposed to the biological environment, blood plasma proteins and ions selectively adsorb on the outer surface of the cell [35]. The complex interface depends on the physical and chemical characteristics of the NPs, as well as the biological characteristics of the environment [36]. In the present study, we observed that TiO_2_ NPs had an average size of 150 nm in the culture medium. We can notice an increase in the average size after the addition of the culture medium due to the adsorption of proteins and ions on the TiO_2_ NPs surface, which can be correlated with the change in surface charge, identified by zeta potential analysis. Furthermore, in our previous study, we confirmed the adsorption of calcium and phosphate on the surface of TiO_2_ NPs, which are important mediators of bone mineralization [5].

To understand the influence of TiO_2_ NPs on bone cell mineralization, we used a mature osteoblast line, cultured both in monolayer (2D) and spheroids (3D), the former characterized previously [2]. Spheroids and monolayer cells were treated with 100 µg/mL of NPs for 21 days. In both models (2D and 3D), we observed the internalization of TiO_2_ NPs in membrane vesicles (with 72 h). Some studies have shown that NPs can be internalized in a dose-dependent manner, accumulating preferentially in the perinuclear region, and having as their final destination the lysosomes [35,37]. Normally TiO_2_ NPs are not observed dispersed in cell cytoplasm [5,35,37]. However, the effect of TiO_2_ NPs on cells is directly related to their size distribution, crystal structure, as well as corona formation [35]. Recently, the formation of a bio-camouflage rich in calcium, phosphorus, and hydroxyapatite crystals around TiO_2_ NPs was demonstrated, which is known to facilitate the internalization in 2D and 3D osteoblastic models since the detected chemical elements are essential for bone cell metabolism and mineralization [2,5,38]. 

The present study demonstrated that TiO_2_ NPs did not alter the viability of osteoblasts in both cell models (with 21 days). Concomitantly, they did not change the osteoblast morphology or spherical shape of the 3D model upon internalization of the NPs. Interestingly, they were able to stimulate an increase in calcium deposition, which is indicative of the activation of a mineralization process in osteoblastic spheroids. In the present study, results of alkaline phosphatase synthesis and calcium labeling demonstrated that TiO_2_ NPs increased osteoblast differentiation that induced greater mineralization in a 3D culture model, suggesting that the 3D architecture possibly increases cell surface interaction with previously reported TiO_2_ NPs bio-camouflaged [2]. The mineralization increase in 3D models after exposure to NPs may be related to the greater cell surface capable of contacting NPs when compared to the monolayer (2D), enhancing the stimulatory effects of TiO_2_ NPs [39]. This is consistent with previous studies that reveal that 3D osteoblasts models when exposed to TiO_2_ NPs, compared to monolayer cells, induce the secretion of vascular endothelial growth factor (VEGF), activating a cascade of events resulting in higher type I collagen production [39,40,41,42,43]. Bone mineralization is the first step for implant osseointegration and begins when collagen I acts as a three-dimensional scaffold for hydroxyapatite deposition [44]. Another study reported greater osteogenic differentiation when using 3D collagen gel culture [34]. Studies also showed that the 2D cell model does not yet seem to be the better model to study interaction with NPs; instead, the spheroids are also promising for application to 3D bioprinting tissue models with biomaterial scaffolds, as an innovative technology to improve bone osteointegration [45].

Unfortunately, there is no consensus in the literature on how to evaluate the biological effect of TiO_2_ NPs. Without standardized protocols to assess the biological impacts of NPs, it is necessary to validate safe assessments and mitigate potential health impacts, moving toward the evaluation and development of new cellular study models to better mimic the biological environment [37]. Although osteoblastic spheroids have their advantages compared to monolayers—such as reproducibility, better nutrients, oxygen diffusion gradients, improved cell–cell interactions, matrix deposition, and models with various cell stages (proliferating, quiescent, apoptotic, hypoxic, and necrotic cells) [46,47], 3D spheroid models have not been validated as realistic in vitro models [29,46,48]. One of the main drawbacks of spheroids is that the porosity and mechanical properties is difficult to be studied. Thus, efforts should be made to improve 3D bone cell models to recapitulate the bone microenvironment that is known to be constituted by different cell types and has dynamic and metabolic activity. 

TiO_2_ NPs released from dental implants are, on the one hand, considered the cause of clinical peri-implant bone loss; on the other hand, they may be able to stimulate the production of a mineralized extracellular matrix in osteoblast spheroids [48]. Another important aspect is that spheroids can respond physiologically better to the stimuli of TiO_2_ NPs, which corroborates the development of new studies to create new models applied in clinical studies, to favor the process of bone remodeling and alternative treatment for periodontitis and peri-implantitis. In addition, the spheroids themselves can be applied to high-cell-density tissue models, innovative technology for bone augmentation, and soft tissue replacement procedures [45]. Therefore, the combination of TiO_2_ NPs with spheroid cells should be an interesting approach for tissue reconstruction.

Lastly, our results demonstrate that TiO_2_ NPs increase calcium deposition in 3D versus 2D cultures. Although this study revealed interesting findings regarding the behavior and role of TiO_2_ NPs in generating stimuli for mineralization in 3D models only, it should be noticed that our results are limited to the conditions tested and the experimental setup. Further studies should be encouraged, and further evaluations using the quantification of genes that act on differentiation and mineralization should be performed. However, our results help to better understand the possible impact of 3D culture in dentistry, and also open a discussion about the dual role of TiO_2_ NPs, which on one side can activate an inflammatory response that leads to bone resorption. However, on the other hand, it is activating mineralization. Our findings are considered clinically relevant, since, for the first time, we report that at the bone-implant interface, TiO_2_ NPs besides the activation of macrophages can also stimulate osteoblasts that play a fundamental role in the mineralization process.

## 5. Conclusions

In this study, the cells were exposed to TiO_2_ NPs at concentrations up to 100 μg/mL in 2D and 3D models for up to 21 days of exposure.

TiO_2_ aggregates were dispersed to nanometric size and characterized successfully. Its internalization in both cell models showed no differences in cell morphology or viability and bone mineralization induction in a dose-dependent form in both culture models. 

However, the mineralization process was more intense in the 3D spheroid culture compared to the 2D monolayer model. 

This brings a new discussion about the possible advantages of TiO_2_ NPs on bone mineralization, which may suggest that the action of nanometric particles can contribute to the osseointegration process in titanium dental implants, reducing periprosthetic failures and using 3D cell models as an innovative technology to improve bone osteointegration induced by nanoparticles.

## Figures and Tables

**Figure 1 nanomaterials-13-00425-f001:**
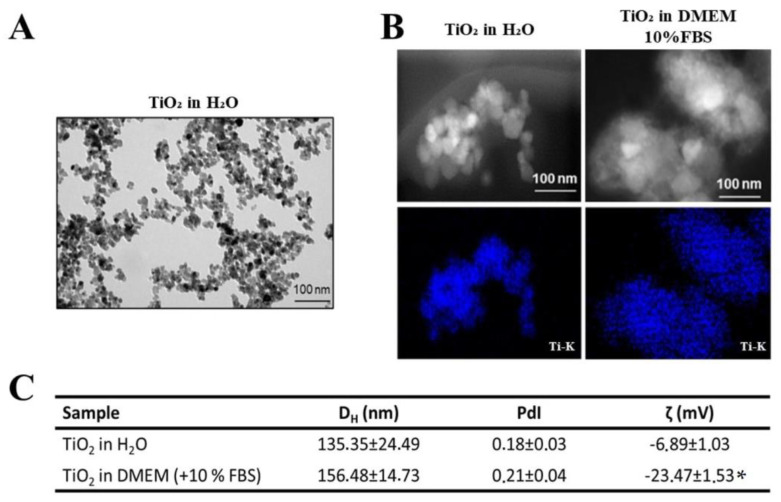
Characterization of TiO_2_ NPs: (**A**) Transmission electron micrographs (TEM) of TiO_2_ NPs in ultrapure water without dispersion. (**B**) Dark-field STEM micrographs of TiO_2_ NPs after dispersion in water and cell culture medium and STEM/EDS Ti-K map confirming titanium presence (in blue). Scale bar: 100 nm. (**C**): Hydrodynamic diameter (D_H_ (nm)) and polydispersity index (PDI) of TiO_2_ NPs after dispersion in ultrapure water and cell culture medium obtained by dynamic light scattering (DLS) and analysis of surface charge by zeta potential of TiO_2_ NPs (ζ (mV)) in water and culture medium. The results represent the mean ± standard deviation of three independent experiments performed in triplicate of measurement (* *p* < 0.05 vs. TiO_2_ NPs in water).

**Figure 2 nanomaterials-13-00425-f002:**
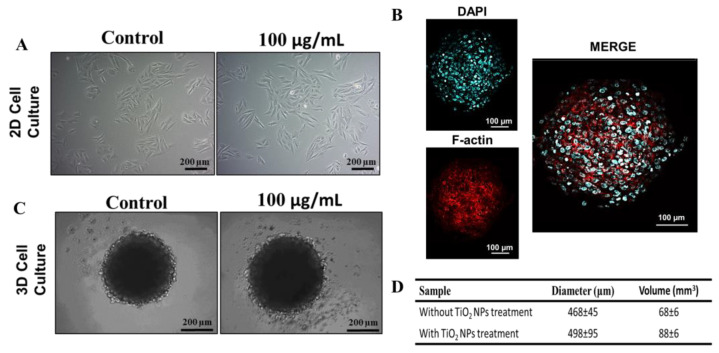
Treatment of TiO_2_ NPs in 2D and 3D cultures of human osteoblasts (SAOS-2): (**A**) Phase contrast micrograph of SAOS-2 in 2D with or without 100 µg/mL of exposition to TiO_2_ NPs for 72 h (scale bar: 200 µm) (**B**) Spheroid cytoskeleton (3D), nucleus in blue (stained with DAPI) and the actin filaments in red (stained for F-actin) (scale bar: 100 µm). (**C**) Phase contrast micrograph of SAOS-2 in 3D with or without 100 µg/mL of exposition to TiO_2_ NPs for 72 h (scale bar: 200 µm). (**D**) Average diameter and volume of spheroids. The baseline condition was used as a control. The results are representative images of three independent experiments.

**Figure 3 nanomaterials-13-00425-f003:**
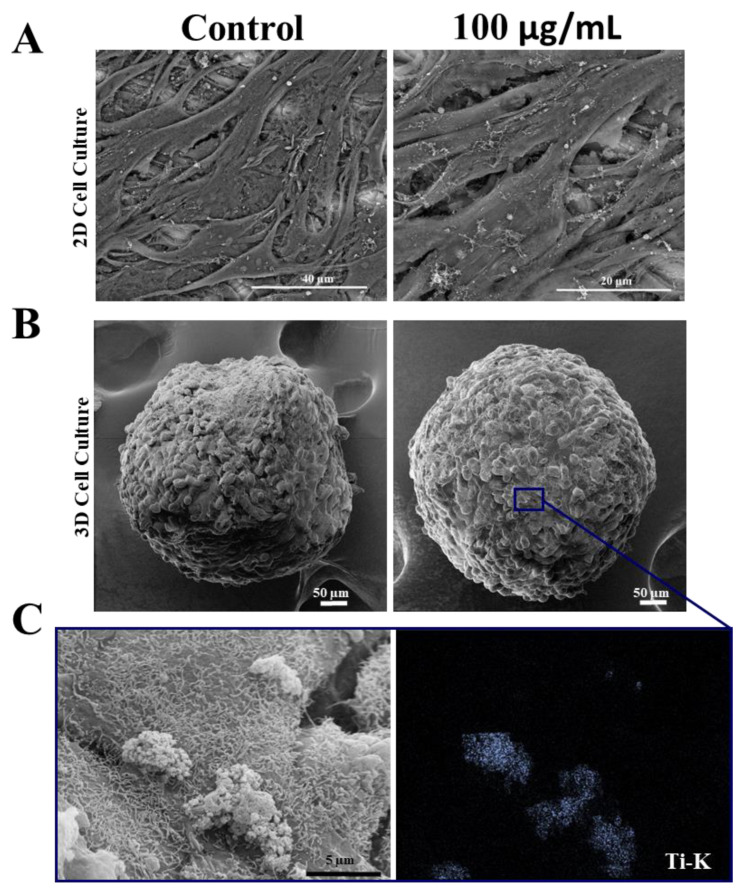
Morphology of human osteoblasts (SAOS-2) cultured in 2D and 3D after exposure to TiO_2_ NPs: (**A**) Scanning electron micrographs showing SAOS-2 cultured in 2D and in spheroids (**B**) after treatment with 100 μg/mL of TiO_2_ NPs for 3 days. (**C**) STEM/EDS Ti-K map analysis. Ti showed in blue point interacting with cells. Control: cultivation without NPs. The results are representative images of three independent experiments. (scale bar: 40 µm, 20 µm, 50 µm and 5 µm).

**Figure 4 nanomaterials-13-00425-f004:**
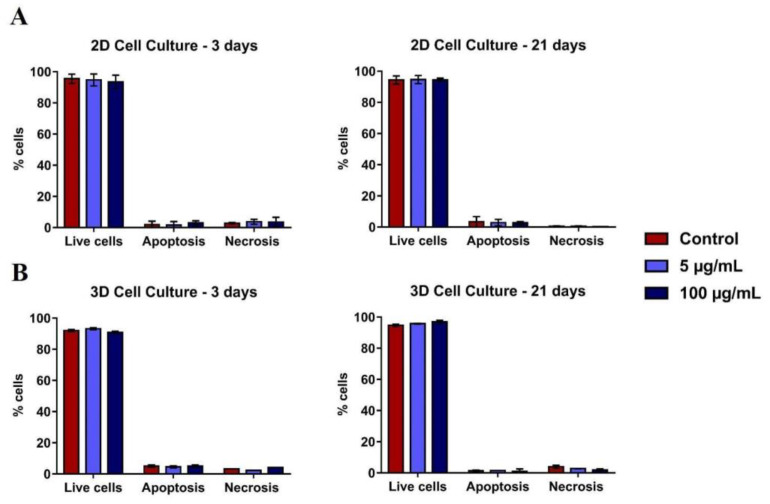
Viability of 2D and 3D human osteoblasts (SAOS-2): SAOS-2 were exposed to TiO_2_ NPs (100 μg/mL) for 3 and 21 days and the PI/Annexin assay was performed by flow cytometry of cells in (**A**) 2D and (**B**) 3D. The baseline condition was used as a control. The results are the mean ± standard deviation of three independent experiments (no statistical difference).

**Figure 5 nanomaterials-13-00425-f005:**
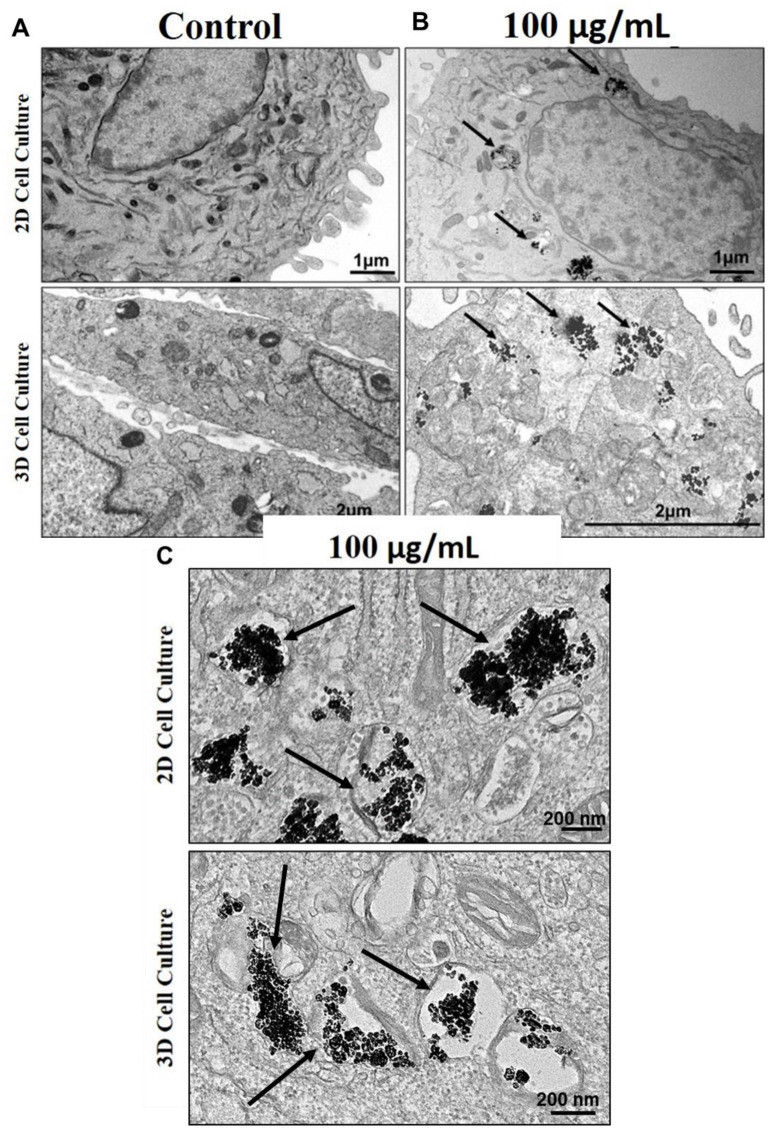
Internalization of TiO_2_ NPs: Transmission electron microscopy (TEM) micrographs showing TiO_2_ NPs internalization in human osteoblasts (SAOS-2) 72 h cultured: (**A**) control, cultured without NPs or (**B**,**C**) treated with 100 µg/mL of TiO_2_ NPs (black arrow). The results are representative images of three independent experiments. (scale bar: 1 µm, 2 µm, and 200 nm).

**Figure 6 nanomaterials-13-00425-f006:**
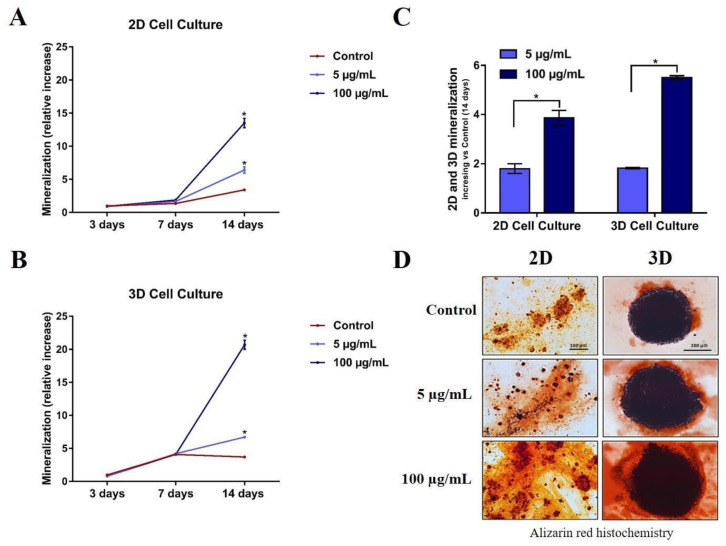
Mineralized matrix produced by human osteoblasts (SAOS-2) in 2D and 3D cells exposed to TiO_2_ NPs (0, 5 and 100 μg/mL) during 3 and 14 days in osteogenic medium. Alizarin red assay shows a dose-dependent matrix production in (**A**) 2D SAOS-2 model and (**B**) 3D SAOS-2 models. The graphics in (**C**) present a comparison between the inorganic matrix production on both 2D and 3D models at 14 days (final experimental time) normalized *vs* each control to show dose-dependent mineralization fold increase in the different models. The results are average ± standard deviation of three independent experiments * *p* < 0.05; (**D**) representative images obtained through optical microscopy.

**Figure 7 nanomaterials-13-00425-f007:**
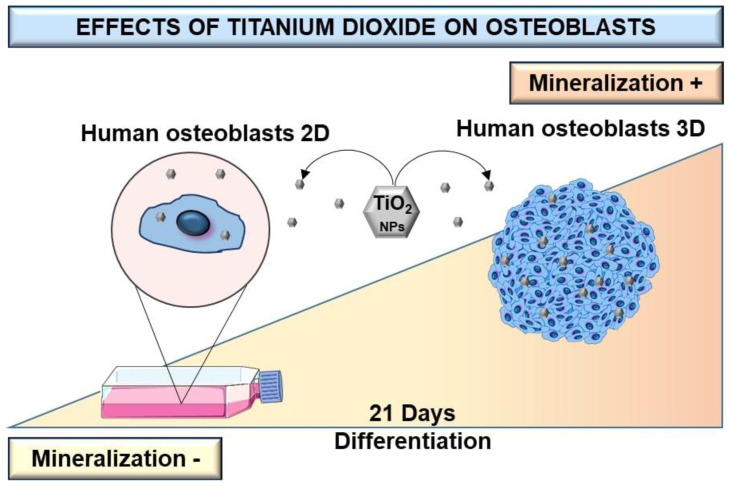
Scheme of differentiation and production of a mineralized matrix of human osteoblasts SAOS-2 cultured in monolayer and 3D spheroid cell models after exposition to TiO_2_. After high TiO_2_ NPs exposure, the dose-dependent increase of bone-mineralized matrix in the 3D cells model is higher than in monolayer (2D) culture.

## Data Availability

The data that support the findings of this study are available from the corresponding author upon reasonable request.

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
