# Peer review of "The Effects of Titanium Dioxide Nanoparticles on Osteoblasts Mineralization: A Comparison between 2D and 3D Cell Culture Models"

_nanomaterials, 2023, doi:10.3390/nano13030425_

Round 1
Reviewer 1 Report
The manuscript is not very interesting, not innovative and too 'medical' with respect to the purpose of the journal.
Need for numerous revisions that denote superficiality and scientific shortcomings in the treatment of the results.
1) NPs of TiO2 are commercial and do not need such important characterisations to be reported in the manuscript.
2) use of plural and singular NPs is always coincidental
3) ml must changed in mL
4) markers Figure 3A wrong, Figure 5 not understood as they are
5) Figure 3C is not in the caption
6)SM material not properly captioned
7) Figure 6 is missing error
Author Response
The authors would like to express their gratitude to the editor and reviewers for considering our manuscript and providing valuable suggestions to improve its quality. A detailed reply to the comments of the reviewers is presented on file.
Best regards.

Reviewer 2 Report
The authors have studied the effects of titanium dioxide nanoparticles on osteoblasts mineralization. A comparison of both 2D and 3D cell culture models have been made. There are various issues that need to be addressed before publication.
1. There are many formatting issues in this paper. What is Z potential and zeta potential? The notations are not consistent.
2. Scale bars are absent in Fig. 2 A and C.
3. The micron markers on Fig. 3A are not correct.
4. Please check the scale bars on Fig. 5.
5. The discussion section is very short.
6. The conclusions are too broad. Important conclusions should be written in bulleted form.
7. Additionally, English should be polished.
Author Response

(The authors gave the same response as above.)

Reviewer 3 Report
I would like to recommend this manuscript for publication after minor revision:
1. Line 78, the “2” in the “TiO2” should be subscript.
2. Line 108, the “4” in the “NH4Cl” should be subscript too.
3. Figure 1, in the legend, (scale bar: 100 nm) which was displayed in the end of the table is not correct. Please move it to the end of Figure 1A or Figure 1B. I didn’t see the “*” mark in the table, so what does the (*p<0.05) mean? Who compared with?
4. The XRD is suggested for characterize the crystal form of TiO2 nanoparticles. Is it anatase, rutile, or both. Because this material characteristic has different effects on cells.
5. The two scalebars of Figure 3A are mistake. Please correct them.
6. Figure 5, in the legend, I can’t see scalebar: 200μm in the figures. Please check the figures.
7. Figure 6, it is very strange that the Figure 6D as a Visual picture has the *p<0.05 in its legend, and This usually appears in the statistical graph. Please carefully check it.
8. A recent reference about biomaterials for Osteoblast is suggested for the Introduction < Jingan Li, Panyu Zhou, Liguo Wang, Yachen Hou, Xueqi Zhang, Shijie Zhu, Shaokang Guan. Investigation of Mg-xLi-Zn alloys for potential application of biodegradable bone implant materials. Journal of Materials Science: Materials in Medicine 2021; 32:43.>.
Author Response

(The authors gave the same response as above.)

Round 2
Reviewer 1 Report
Accepted
Reviewer 2 Report
The authors have revised the manuscript perfectly.
Reviewer 3 Report
The manuscript has been well revised. I recommend it for publication.